# Comparison of Image Quality and Quantification Parameters between Q.Clear and OSEM Reconstruction Methods on FDG-PET/CT Images in Patients with Metastatic Breast Cancer

**DOI:** 10.3390/jimaging9030065

**Published:** 2023-03-09

**Authors:** Mohammad Naghavi-Behzad, Marianne Vogsen, Oke Gerke, Sara Elisabeth Dahlsgaard-Wallenius, Henriette Juel Nissen, Nick Møldrup Jakobsen, Poul-Erik Braad, Mie Holm Vilstrup, Paul Deak, Malene Grubbe Hildebrandt, Thomas Lund Andersen

**Affiliations:** 1Department of Clinical Research, University of Southern Denmark, 5000 Odense, Denmarkthomas.lund.andersen@regionh.dk (T.L.A.); 2Department of Nuclear Medicine, Odense University Hospital, 5000 Odense, Denmark; 3Centre for Personalized Response Monitoring in Oncology, Odense University Hospital, 5000 Odense, Denmark; 4Department of Oncology, Odense University Hospital, 5000 Odense, Denmark; 5Department at Clinical Engineering, Region of Southern Denmark, 6200 Aabenraa, Denmark; 6Healthcare Science Technology, GE Healthcare, Chicago, IL 06828, USA; 7Centre for Innovative Medical Technology, Odense University Hospital, 5000 Odense, Denmark; 8Department of Clinical Physiology and Nuclear Medicine, Rigshospitalet, 2100 Copenhagen, Denmark

**Keywords:** reconstruction algorithm, FDG-PET/CT, metastatic breast cancer, Q.Clear

## Abstract

We compared the image quality and quantification parameters through bayesian penalized likelihood reconstruction algorithm (Q.Clear) and ordered subset expectation maximization (OSEM) algorithm for 2-[^18^F]FDG-PET/CT scans performed for response monitoring in patients with metastatic breast cancer in prospective setting. We included 37 metastatic breast cancer patients diagnosed and monitored with 2-[^18^F]FDG-PET/CT at Odense University Hospital (Denmark). A total of 100 scans were analyzed blinded toward Q.Clear and OSEM reconstruction algorithms regarding image quality parameters (noise, sharpness, contrast, diagnostic confidence, artefacts, and blotchy appearance) using a five-point scale. The hottest lesion was selected in scans with measurable disease, considering the same volume of interest in both reconstruction methods. SUL_peak_ (g/mL) and SUV_max_ (g/mL) were compared for the same hottest lesion. There was no significant difference regarding noise, diagnostic confidence, and artefacts within reconstruction methods; Q.Clear had significantly better sharpness (*p* < 0.001) and contrast (*p* = 0.001) than the OSEM reconstruction, while the OSEM reconstruction had significantly less blotchy appearance compared with Q.Clear reconstruction (*p* < 0.001). Quantitative analysis on 75/100 scans indicated that Q.Clear reconstruction had significantly higher SUL_peak_ (5.33 ± 2.8 vs. 4.85 ± 2.5, *p* < 0.001) and SUV_max_ (8.27 ± 4.8 vs. 6.90 ± 3.8, *p* < 0.001) compared with OSEM reconstruction. In conclusion, Q.Clear reconstruction revealed better sharpness, better contrast, higher SUV_max_, and higher SUL_peak_, while OSEM reconstruction had less blotchy appearance.

## 1. Introduction

Positron emission tomography with integrated computed tomography (PET/CT) is broadly used in the initial diagnosis, staging, and therapeutic response evaluation of numerous malignant diseases [1]. There are continuous technical improvements in PET/CT scanners, leading to improved imaging quality from developed hardware specifications and reconstruction algorithms [2,3,4]. Novel PET/CT scanners based on digital silicon photomultiplier (SiPM) technology have become the new standard in PET by replacing the older generation of PET/CT systems based on analog photomultiplier tubes. This leads to a considerable improvement in image contrast and noise level [5,6], which could provide better diagnostic accuracy and overall image quality, compared with analog PET/CT scanners [3,4,7,8]. A new reconstruction algorithm employing the block sequential regularized expectation maximization (BSREM) technique under the commercial name Q.Clear has been introduced. This method allows for fully convergent iterative reconstruction, resulting in higher image contrast while suppressing noise compared with ordered subset expectation maximization (OSEM) [9]. Advanced reconstruction methods aim to improve not only the quality of imaging but also quantitative measures [7], as using the Q.Clear algorithm potentially increases the maximal standardized uptake (SUV_max_) values within metastatic lesions compared with OSEM reconstruction [10].

2-deoxy-2-[^18^F]fluoro-D-glucose PET/CT (2-[^18^F]FDG-PET/CT) has increasingly been introduced in metastatic breast cancer based on excellent sensitivity (over 95%) with regard to detection of distant metastases [11]. Response evaluation using 2-[^18^F]FDG-PET/CT may improve clinical management and survival [12,13,14]. Quantitative PET/CT is becoming increasingly important for more objective evaluations of tumor response [7], with the PET Response Criteria in Solid Tumors (PERCIST) being suggested as feasible and valuable criteria in breast cancer [15,16,17].

Prior studies have indicated Q.Clear reconstruction to be superior in overall image quality, improved contrast recovery and noise suppression, better lesion detectability, and more accurate quantification compared with OSEM reconstruction [18,19,20]. Literature comparing the two reconstruction methods in a clinical setting is lacking and is warranted before introducing the Q.Clear algorithm as a reconstruction method for semi-quantitative PERCIST analyses in clinical practice [21]. Thus, we aimed to compare the two reconstruction algorithms (Q.Clear vs. OSEM) regarding the overall image quality and quantification parameters in a clinical setting of patients being response-evaluated during treatment for metastatic breast cancer.

## 2. Materials and Methods

### 2.1. Patients

A prospective comparative study was conducted at the Department of Nuclear Medicine at Odense University Hospital (Denmark), following the STROBE guideline [22]. Patients in the present study represent a subpopulation of a patient group analyzed in a larger prospective study of response monitoring in metastatic breast cancer (Clinical.Trials.gov: NCT03358589). Women referred to the Department of Oncology (Odense University Hospital) with advanced breast cancer between 2018 (September) and 2020 (September) and examined initially with 2-[^18^F]FDG-PET/CT were considered eligible. The Danish Ethics Committee approved the study protocol (S-20170019), and all subjects signed written informed consent. The research was conducted in accordance with the Declaration of Helsinki, and all the scans were performed following the guideline of European Association of Nuclear Medicine (EANM) [23].

Inclusion criteria for NCT03358589 were biopsy-verified relapsed or de novo metastatic breast cancer (biopsy verification of primary tumor and disseminated disease at baseline scan), imaging performed on PET/CT scanners with digital technology, and available clinico-histopathological data. Exclusion criteria were age less than 18 years and patients in treatment for other invasive cancers [24].

The baseline scans were performed prior to the treatment initiation; patients were monitored on the same PET/CT scanner and were scanned according to the standardization criteria suggested by PERCIST [17]. Patients were scanned with 9–12 weeks of imaging intervals according to Danish clinical guidelines. The baseline, first follow-up, and second follow-up scans were included in the present analysis. Scans were analyzed with both Q.Clear and OSEM reconstruction algorithms by the same group of experienced nuclear medicine physicians comparing overall image quality parameters, maximum standardized uptake value (SUV_max_), and peak lean body mass corrected SUV (SUL_peak_).

### 2.2. PET/CT Imaging Protocol

PET/CT data were acquired on GE Discovery MI 4-ring PET/CT (GE Healthcare, Waukesha, WI, USA) scanners with a field of view of 25 cm. PET scans were performed 60 min after injection of 4 mBq/kg FDG (min 200 MBq, max 400 MBq) using a standard whole-body (head-to-thigh) acquisition protocol, with slice overlaps of 40% and acquisition time of 1.5 min per bed position. PET datasets were reconstructed using time-of-flight 3D OSEM (GE VPFX, 4 iterations, 17 subsets) with point-spread-function correction (GE SharpIR) and using Q.Clear (β = 250) in matrix sizes of 256 × 256 (pixel size 2.74 mm). Corrections for attenuation, scatter, randoms, deadtime, and normalization were performed inside the iterative loop. Attenuation correction was based on a dedicated helical CT attenuation correction scan acquired after the PET scan using a standard CT protocol with a scan field of view of 70 cm. Data were reconstructed with a standard filter into trans-axial slices with a field of view of 50 cm, matrix size of 512 × 512 (pixel size 0.98 mm), and a slice thickness of 3.75 mm [23].

### 2.3. Qualitative Image Analysis

A clarification session was held by a senior nuclear medicine specialist (M.H.V.), ensuring the use of the same approach by the interpreters before analyzing the image quality. Three experienced nuclear medicine physicians analyzed the image quality parameters on both Q.Clear and OSEM reconstruction algorithms. One physician evaluated each scan regarding the qualitative image analyses. The same physician performed the analyses for two reconstruction methods at the same time on two separate screens (side-by-side) regarding six image quality parameters, while they were blinded to the reconstruction methods. Using a five-point scale (1 = worst and 5 = best, Table 1), the following quality parameters were compared between the reconstruction methods: noise, sharpness, contrast, diagnostic confidence, artefacts, and blotchy appearance [25]. The parameters were evaluated on a subjective scale, while the physicians were experienced enough to be good at assessing noise of the scans, as well as separating the parameters with overlap such as noise, contrast, and sharpness. The interpreters evaluated the scans independently of one another, and they had knowledge of the clinical indication of PET/CT.

### 2.4. Quantitative Image Analysis

PET/CT scans were evaluated for quantitative measurements. The hottest lesion according to one-lesion PERCIST was selected in scans with measurable disease, considering the same image number, considering the same volume of interest (VOI), and using the PETVCAR automatic software (AW version 3.2, GE Healthcare, Chicago, IL, USA) in both reconstruction methods. SUL_peak_ was defined as the highest possible mean value of a 1 cm^3^ spherical in the VOI positioned within the metastatic lesions. Patients were checked for serum glucose level, ensuring that it was in an acceptable range (less than 200 mg/dL) according to the PERCIST criteria [17]. SUV_max_ was defined as the maximum uptake in the VOI that reflects the maximum tissue concentration of FDG uptake in the tumor. Body weight and height were used for SUV_max_ normalization [26]. SUL_peak_ (g/mL) and SUV_max_ (g/mL) were calculated and compared for the same hottest lesion. Quantification of FDG uptake was performed on scans with measurable disease at baseline and for comparable scans during follow-up (i.e., SUL_peak_ and SUV_max_ were not measured for scans with complete metabolic response).

### 2.5. Outcome Measure and Statistical Analysis

Continuous data were presented using the median (range) and mean ± standard deviation. Frequencies and respective percentages were given for categorical variables. A t-test was used to compare the six parameters regarding the image quality and quantitative parameters (SUL_peak_ and SUV_max_) of the hottest lesion between the two algorithms. The statistical level of significance was set to 0.05. All statistical analyses were conducted with STATA/IC (version 16.1, StataCorp, College Station, USA).

## 3. Results

A total of 37 patients with 37 baseline scans and 63 follow-up scans (including first and second follow-up scans) were available for the analysis. A study flowchart is seen in Figure 1. The clinical and histopathological information of included patients is summarized in Table 2. More detailed information of included patients is available in a previous publication [24].

Comparing the parameters related to the quality of images, Q.Clear had significantly better sharpness (mean scores of 4.65 vs. 3.91) and contrast (mean scores of 4.23 vs. 4.10) compared with the OSEM reconstruction (*p* < 0.001 and *p* = 0.001, respectively), while there was no significant difference regarding noise, diagnostic confidence, and artefacts when comparing the two reconstruction methods. The OSEM reconstruction had less blotchy appearance (4.57 vs. 4.34) compared with Q.Clear reconstruction (*p* < 0.001). Scores related to imaging quality parameters are summarized in Table 3. An example of 2-[^18^F]FDG-PET/CT, comparing the sharpness and contrast using OSEM and Q.Clear reconstructions, respectively, is shown in Figure 2.

A total of 31/37 (84%) patients had measurable disease at baseline (Figure 1), for whom quantitative analysis was performed on the hottest lesion according to the PERCIST criteria. At follow-up scans, quantitative analyses were performed in 44/63 (70%) scans being comparable according to the PERCIST criteria. Q.Clear reconstruction had significantly higher SUL_peak_ (5.33 ± 2.8 vs. 4.85 ± 2.5, *p* < 0.001) and SUV_max_ (8.27 ± 4.8 vs. 6.90 ± 3.8, *p* < 0.001) compared with the OSEM reconstruction (Table 4). When comparing the two reconstruction methods for change in SUL_peak_ and SUV_max_ between the two following scans, there was no significant difference in the median SUL_peak_ changes, while the median SUV_max_ changes were significantly higher for Q.Clear reconstruction.

## 4. Discussion

A prospective comparison of OSEM and Q.Clear reconstruction algorithms was conducted on image quality parameters and quantitative analysis on 2-[^18^F]FDG-PET/CT scans of patients with metastatic breast cancer. The Q.Clear reconstruction showed significantly improved sharpness and contrast compared with an OSEM reconstruction, while OSEM reconstruction had a less “blotchy appearance” compared with the Q.Clear reconstruction, which may be a consequence of reduced noise in OSEM reconstruction. Registered SUL_peak_ and SUV_max_ values with Q.Clear reconstruction were significantly higher than with OSEM reconstruction. Changes in SUL_peak_ values over the follow-up period stayed independent of the reconstruction method, while the changes related to SUV_max_ were significantly higher in Q.Clear reconstruction.

According to our study, images performed with Q.Clear reconstruction had better contrast and sharpness than images from OSEM reconstruction, which is in line with the results of other studies indicating a superiority for the reconstruction methods based on the BSREM technique over the OSEM reconstruction [27,28]. A more “blotchy appearance” on Q.Clear images could be a consequence of a missing Gaussian filter, while using a 4 mm Gaussian filter regularly in OSEM reconstruction smoothed the blotchy appearance and resulted in a better image quality [28]. We found no difference in the diagnostic confidence between the two methods, indicating that any of the reconstruction methods may be preferred for clinical application.

Similar studies have also reported better overall image quality by Q.Clear compared with OSEM in [^68^Ga]Ga-DOTANOC PET/CT scans [29], ^18^F-fluciclovine PET/CT scans [20], and [^68^Ga]Ga-PSMA PET/CT scans [30]. This indicates that the improved image quality provided by Q.Clear is not exclusively dedicated to 2-[^18^F]FDG-PET/CT scans. Furthermore, a few studies have reported that Q.Clear had better image quality than OSEM in PET/MR scans [2,18,31]. The reason could be the same as PET/CT scans, presuming the inability of OSEM reconstruction to achieve full convergence due to increased noise and the iteration times. However, more studies including phantom data are needed to ensure the preferred β value of reconstructions ensuring the optimal image quality [20,29,30]. The reason for the better image quality with the Q.Clear algorithm could be that Q.Clear reduces the noise by resembling an adaptive filter with adjustable filter width and improving the contrast by increasing quantification, effectively creating the effect of better image quality. This is in line with our results, indicating improved contrast on Q.Clear reconstructions, which allows reaching full convergence without the excessive typical noise of OSEM [29]. This effectively limits the number of iterations of the OSEM algorithm to avoid excessive noise within the image, resulting in a lack of convergence and decreasing image contrast [2]. On the other hand, Q.Clear can achieve full convergence, resulting in higher resolution and more precise quantitative measurements due to the noise regularization [19,32].

There was a significant difference between the absolute values of SUV_max_ and SUL_peak_ within the OSEM and Q.Clear reconstruction algorithms, which is in line with the results of previous studies indicating that Q.Clear allows a significant increase in quantitative parameters and better reflects the true uptake [9,20]. Therefore, Q.Clear has the potential to provide an improved quantification accuracy, which could be beneficial for research purposes, as well as using quantification-based analyses (e.g., PERCIST analysis) in clinical practice.

Lundeberg et al. compared the two reconstruction modalities on lung cancer patients in a clinical setting and reached the similar results indicating that the Q.Clear reconstruction provides a higher SUV_max_ for suspected lymph node metastases compared to the OSEM reconstruction. However, a higher values of SUV_max_ did not lead to an improvement in detection of metastatic lesions [33]. In addition, our results showed that SUL_peak_ changes over the follow-up period were not associated with the reconstruction algorithm, as opposed to SUV_max_, for which changes were higher for the Q.Clear than OSEM reconstruction, indicating SUL_peak_ to be robust for PERCIST analysis across reconstruction algorithms. This complies with the proposed upward bias for SUV_max_ using a single pixel, as the size of a single voxel may differ considerably among PET/CT scans resulting in different noise levels in the metric followed by various filtering, while PERCIST suggests using a larger region of interest by SUL_peak_ [17]. Q.Clear also showed a clinically relevant recovery coefficient for different sphere diameters (10–37 mm), which is beneficial for lesion detection and is more compatible with the quantification of lesions according to PERCIST criteria [34].

It has also been reported that PERCIST has high applicability [16], has a higher level of overall interrater agreement and reliability compared with a qualitative assessment [35,36], and is superior in the detection of new lesions or unequivocal progression in nontarget lesions [37]. Therefore, clinical implementation of the PERCIST assessment may improve the prognostic stratification [15,37,38] and provide a standardized approach independent of interpreters and reconstruction methods. The difference in the absolute value of SUL_peak_ may result in dissimilar response categories according to the PERCIST criteria and eventually affect the patients’ treatment plan. Hence, a clinical indication of the Q.Clear reconstruction algorithm may lead to a more precise treatment monitoring by PET/CT through an improved quantification accuracy [39]. However, the clinical indication of the Q.Clear algorithm in the treatment evaluation of lymphoma patients is reported to be uncertain, which could be explained by the incompatibility with the current guidelines [10]. Therefore, compatibility of Q.Clear with existing guidelines is required before introducing the algorithm into clinical practice to have a significant impact on patient management.

A strength of this study was the inclusion of clinical follow-up of metastatic patients representing daily clinical practice. Furthermore, the PERCIST criteria were used for response monitoring and quantification of FDG uptake with strict compliance with standardization criteria such as the PERCIST guidelines [17]. Accordingly, patients were followed on the same type of scanner, eliminating scanner variation effects, while comparing quantitative measures between follow-up scans. The interpreters for imaging quality parameters were blinded to the reconstruction methods, and the same physician evaluated both reconstruction methods at the same time (side-by-side). As a limitation, we only performed a visual comparison in terms of overall image quality, which could be biased by the personal preferences of the interpreter physician, as only one operator analyzed each scan. Furthermore, the number and size of the detected lesions for the two reconstruction methods were not measured, which could have strengthened the results of quantitative analyses.

Future multicenter studies on a larger number of scans evaluated by several experienced nuclear medicine physicians could verify the results of the current study. The lesion-based accuracy for the two reconstruction methods and the potential effect on response categories remain unanswered. Furthermore, the determination of the optimal penalization factor (β-value) for clinical use and phantom measurements related to Q.Clear should be considered in future studies.

## 5. Conclusions

Q.Clear reconstruction showed a significantly better sharpness and contrast compared with OSEM reconstruction, while the blotchy appearance was less evident in OSEM reconstruction. There was no difference in diagnostic confidence between the two reconstruction algorithms, making them equally perfect for daily clinical practice. The Q.Clear algorithm had higher quantitative measures with higher SUV_max_ and SUL_peak_ than OSEM reconstruction. SUL_peak_ changes at follow-up scans stayed independent of the reconstruction methods, indicating SUL_peak_ to be robust for PERCIST analyses regardless of the reconstruction method.

## Figures and Tables

**Figure 1 jimaging-09-00065-f001:**
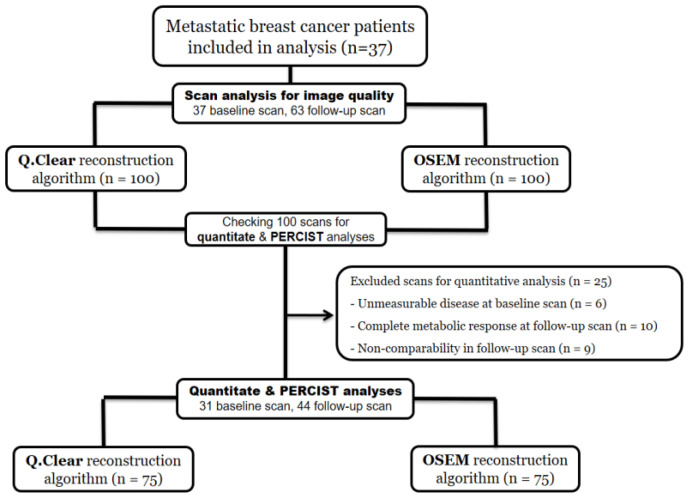
Study flowchart for image quality and quantitative analyses (OSEM: ordered subset expectation maximization; Q.Clear: Refers to the reconstruction algorithm using block sequential regularized expectation maximization; PERCIST: PET Response Criteria in Solid Tumors).

**Figure 2 jimaging-09-00065-f002:**
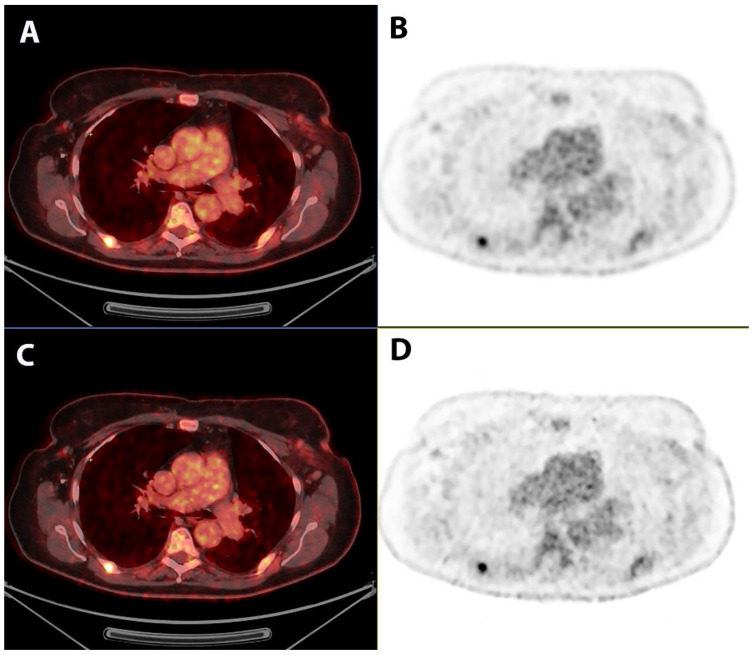
2-[^18^F]FDG-PET/CT scan for a patient with metastatic breast cancer, illustrating the sharpness and contrast via OSEM and Q.Clear reconstructions ((**A**,**B**) OSEM reconstruction vs. (**C**,**D**) Q.Clear reconstruction).

**Table 1 jimaging-09-00065-t001:** Grading scale for subjective image quality evaluation.

Parameters	5	4	3	2	1
Noise	Minimal or no noise	No significant noise	Noisy diagnostic	Significant noise (affects diagnosis)	High-level noise (nondiagnostic)
Sharpness	Excellent sharpness	Good sharpness	Moderate sharpness	Poor sharpness (bad visibility)	Zero visibility (nondiagnostic)
Contrast	Excellent contrast	Very good contrast	Good contrast	Poor contrast (unsatisfactory visualization)	Image similar to use of no contrast (nondiagnostic)
Diagnostic confidence	Completely confidence	High confidence	Good confidence	Poor confidence	No diagnostic confidence (unacceptable)
Artefacts	No artefacts	Insignificant artefacts	Minor artefacts	Major artefacts (diagnosis still possible)	Artefacts affecting diagnostic information
Blotchy appearance	Absent	Mild	Moderate	Significant (diagnosis still possible)	Intense (affecting diagnosis)

**Table 2 jimaging-09-00065-t002:** Clinicopathological characteristics of included patients with metastatic breast cancer.

Characteristics		Results *
Age (years)		71.9 (45.9–91.1)
Primary cancer treatment	Postoperative adjuvant treatment	24 (64.7)
	Adjuvant and neoadjuvant treatments	3 (8.1)
	No treatment/unknown	10 (27.0)
	History of radiotherapy	24 (64.7)
Primary disseminated cancer		12 (32.4)
Histopathology	Adenocarcinoma	28 (75.7)
	Invasive ductal carcinoma	5 (13.5)
	Invasive lobular carcinoma	4 (10.8)
Positive estrogen receptor		32 (86.5)
Negative Herceptin receptor		34 (91.9)
Origin of biopsy **	Bone	13 (35.1)
	Liver	7 (18.9)
	Lung	1 (2.7)
	Lymph nodes	6 (16.2)
	Breast	10 (27.0)
First-line treatment	Endocrine therapy	5 (13.5)
	Endocrine therapy + CDK4/6 inhibitor	24 (64.9)
	Chemotherapy	4 (10.8)
	Others	4 (10.8)

* Data are shown as median (range) and frequency (%). ** In some cases of de novo cancer, we used information from the initial breast biopsy in case of absence of biopsy from metastases.

**Table 3 jimaging-09-00065-t003:** Scores of image quality parameters within the OSEM and Q.Clear reconstruction methods.

Characteristics	OSEM *	Q.Clear *	Mean Difference (95% CI)	*p*-Value
Noise	4.41 ± 0.55	4.42 ± 0.54	0.01 (−0.16–0.14)	0.88
Sharpness	3.91 ± 0.49	4.65 ± 0.59	−0.74 (−0.83–−0.65)	<0.001
Contrast	4.1 ± 0.66	4.23 ± 0.74	−0.13 (−0.22–−0.04)	0.001
Diagnostic confidence	4.52 ± 0.70	4.52 ± 0.69	0 (−0.28–0.28)	0.99
Artifacts	4.37 ± 0.68	4.38 ± 0.66	−0.01 (−0.3–0.01)	0.32
Blotchy appearance	4.57 ± 0.57	4.34 ± 0.59	0.23 (0.12–0.34)	<0.001

OSEM: ordered subset expectation maximization; CI: confidence interval; Q.Clear: refers to the reconstruction algorithm using block sequential regularized expectation maximization (BSREM). * Image quality scores (mean ± standard deviation) are reported using a five-scale questionnaire (1 = worst and 5 = best).

**Table 4 jimaging-09-00065-t004:** Quantitative analysis within the hottest lesion via OSEM and Q.Clear reconstruction methods.

Characteristics		OSEM *	Q.Clear *	Mean Difference (95% CI)	*p*-Value
SUL_peak_	Baseline scans	5.82 (1.4–12.12)	6.84 (1.61–12.95)	−0.6 (−0.82–−0.39)	<0.001
	Follow-up scans	3.01 (1.65–11.01)	3.47 (1.79–12.82)	−0.39 (−0.52–−0.26)	0.001
	All scans	4.3 (1.4–12.12)	4.63 (1.61–12.95)	−0.47 (−0.59–−0.36)	<0.001
	Change to 1st follow-up	1.94 (0.07–5.71)	1.95 (0–5.86)	0.04 (−0.24–0.33)	0.75
	Change to 2nd follow-up	0.53 (0.02–4)	0.8 (0.05–4.16)	0.12 (−0.23–0.26)	0.1
SUV_max_	Baseline scans	8.12 (2.0–18.42)	9.46 (2.37–24.86)	−1.49 (−1.97–1.01)	<0.001
	Follow-up scans	4.61 (2.22–18.42)	5.48 (2.47–24.86)	−1.25 (−1.73–−0.77)	0.005
	All scans	6.16 (2.0–18.42)	7.15 (2.37–24.86)	−1.35 (−1.69–−1.01)	<0.001
	Change to 1st follow-up	2.34 (0–8.3)	2.39 (0–9.5)	0.45 (0.10–0.79)	0.01
	Change to 2nd follow-up	1.1 (0.05–4.92)	1.5 (0.26–11.2)	0.89 (0.20–1.58)	0.04

OSEM: ordered subset expectation maximization; CI: confidence interval; Q.Clear: refers to the reconstruction algorithm using block sequential regularized expectation maximization (BSREM); SUV_max_: maximum standardized uptake value; SUL_peak_: peak lean body mass corrected SUV. * Data are shown as the median (range).

## Data Availability

The datasets analyzed during the current study are available from the corresponding author on reasonable request.

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
