# Peer review of "Comparison of Image Quality and Quantification Parameters between Q.Clear and OSEM Reconstruction Methods on FDG-PET/CT Images in Patients with Metastatic Breast Cancer"

_2313-433X, 2023, doi:10.3390/jimaging9030065_

Round 1

Reviewer 1 Report

The authors designed a manuscript of great quality, presenting an interesting subject, with great prospects of applicability in clinical practice, as the chosen method of image reconstruction might affect the quantitative and qualitative assessment of PET images in patients diagnosed with breast cancer, further affecting their stadialization and treatment. I only have two suggestions for some minor changes:

  1. Was LBM (lean body mass) the method of SUV normalization for both SUVmax and SUVpeak? The authors specified this method only for SUVpeak (SULpeak). If this method was also used for SUVmax, please specifiy this in the Materials and Methods section. If the authors used other normalization methods (i.e. body weight), please do mention.
  2. How many operators performed the quantitative image analysis? If there was more than one operator, was there a good inter-operator reproducibility, especially in the case of SULpeak? Was the segmentation of SULpeak done automatically by the software or manually by operators? This matter is of interest and should, as the authors mentioned the possibility of operator-dependent bias.

In my opinion, other than these two suggestions, the authors managed to cover the subject in its entirety, producing a paper of great value.

Author Response

Dear respected reviewer

Thanks for your constructive comments/suggestions, which were considered in revised version of manuscript.

  1. Was LBM (lean body mass) the method of SUV normalization for both SUVmax and SUVpeak? The authors specified this method only for SUVpeak (SULpeak). If this method was also used for SUVmax, please specify this in the Materials and Methods section. If the authors used other normalization methods (i.e. body weight), please do mention.

Thanks for the comment. Lean body mass was only used for SULpeak, while body weight and height were used for SUVmax normalization. It worth to mentioned that the blood sugar of patients were checked before the scans, just to make sure that they are within the acceptable range according to the PERCIST criteria (less than 200 mg/dl or 11.1 mmol/l). We have added a paragraph to the methods section  to clarify this point.

  1. How many operators performed the quantitative image analysis? If there was more than one operator, was there a good inter-operator reproducibility, especially in the case of SULpeak? Was the segmentation of SULpeak done automatically by the software or manually by operators? This matter is of interest and should, as the authors mentioned the possibility of operator-dependent bias.

Thanks for the comment. For qualitative image analyses, one nuclear medicine physician analyzed each scan with both reconstruction algorithms. For quantitative parameters, again there was only one operator using PETVCAR automatic software in both reconstruction methods. The same volume of interest (VOI) and image number were considered in both reconstruction methods, ensuring a correct comparison for the quantitative analyses (SULpeak) within the methods. Therefore, operator-dependent bias was mainly limited to the qualitative image analyses. We have add a clarification to the methods and discussion section.

Reviewer 2 Report

In the manuscript, "Comparison of image quality and quantification parameters between Q.Clear and OSEM reconstruction methods on FDG-PET/CT images in patients with metastatic breast cancer" authors compare image quality between two PET/CT reconstruction algorithms in a prospective setting.

The topic is relevant to the field, as technical adequacy of reconstruction algorithms is mostly evaluated using phantoms or retrospectively, and prospective clinical trials provide valuable insights. Compared to other published material, the study demographic is original with biopsy-verified relapsed or de novo metastatic breast cancer.

The conclusions are consistent with the study goal and methods. The references are appropriate.

Detailed comments and suggestions are provided below:

Abstract: no comments.
Introduction: no comments.
Materials and methods:
- consider describing each parameter for subjective grading in more detail (Could some of them be redundant?), possibly provide clinical examples;
- consider describing study evaluation methodology in more detail (Did every physician evaluate all scans in both reconstructions in a single seating? If yes, why wasn't interobserver agreement assessed?).
Results: no comments.
Conclusion: no comments.
Tables: no comments.

Figures: consider adding examples with significantly different image quality between the two algorithms.

Author Response

Dear respected reviewer

Thanks for your constructive comments/suggestions, which were considered in revised version of manuscript.

  1. Consider describing each parameter for subjective grading in more detail (Could some of them be redundant?), possibly provide clinical examples.

Thank you for the valuable point. The subjective grading was performed by experienced nuclear medicine physicians trained on data from a range of different scanners and hence noise characteristics giving a basis for image features description. We appreciate that there is a potential correlation between parameters used to describe the images but still believe that they illustrate difference characteristics of the images and are clinically diagnostically relevant. We have made it clear in methods section.

  1. Consider describing study evaluation methodology in more detail (Did every physician evaluate all scans in both reconstructions in a single seating? If yes, why wasn't interobserver agreement assessed?)

Thanks for the comment. Each scan was analyzed by one nuclear medicine physician with both reconstruction methods in same settings. The analyses were done at the same time on two screen (side by side), while the physician was blinded to the reconstruction method. We have add a clarification to the methods section

  1. Consider adding examples with significantly different image quality between the two algorithms.

Thanks for the suggestion. We already inserted an example of the difference between the reconstruction methods regarding the image quality (Figure 2).